# Effects of Fermentation on Bioactivity and the Composition of Polyphenols Contained in Polyphenol-Rich Foods: A Review

**DOI:** 10.3390/foods12173315

**Published:** 2023-09-03

**Authors:** Fan Yang, Chao Chen, Derang Ni, Yubo Yang, Jinhu Tian, Yuanyi Li, Shiguo Chen, Xingqian Ye, Li Wang

**Affiliations:** 1Moutai Group, Institute of Science and Technology, Zunyi 564501, China; 2Key Laboratory of Industrial Microbial Resources Development, Kweichow Moutai Co., Ltd., Renhuai 564501, China; 3Department of Food Science and Nutrition, Zhejiang University Zhongyuan Institute, Zhengzhou 450000, China; 4The Rural Development Academy, Zhejiang University, Hangzhou 310058, China; 5National-Local Joint Engineering Laboratory of Intelligent Food Technology and Equipment, College of Biosystems Engineering and Food Science, Zhejiang University, Hangzhou 310058, China; 6Fuli Institute of Food Science, Zhejiang University, Hangzhou 310058, China

**Keywords:** fermented food, polyphenols, polyphenol-associated enzymes, bioactivity, microorganisms

## Abstract

Polyphenols, as common components with various functional activities in plants, have become a research hotspot. However, researchers have found that the bioavailability and bioactivity of plant polyphenols is generally low because they are usually in the form of tannins, anthocyanins and glycosides. Polyphenol-rich fermented foods (PFFs) are reported to have better bioavailability and bioactivity than polyphenol-rich foods, because polyphenols are used as substrates during food fermentation and are hydrolyzed into smaller phenolic compounds (such as quercetin, kaempferol, gallic acid, ellagic acid, etc.) with higher bioactivity and bioavailability by polyphenol-associated enzymes (PAEs, e.g., tannases, esterases, phenolic acid decarboxylases and glycosidases). Biotransformation pathways of different polyphenols by PAEs secreted by different microorganisms are different. Meanwhile, polyphenols could also promote the growth of beneficial bacteria during the fermentation process while inhibiting the growth of pathogenic bacteria. Therefore, during the fermentation of PFFs, there must be an interactive relationship between polyphenols and microorganisms. The present study is an integration and analysis of the interaction mechanism between PFFs and microorganisms and is systematically elaborated. The present study will provide some new insights to explore the bioavailability and bioactivity of polyphenol-rich foods and greater exploitation of the availability of functional components (such as polyphenols) in plant-derived foods.

## 1. Introduction

Fermented food has been an essential part of the human diet for thousands of years. In the past, fermentation was commonly applied to extend the shelf life of foods and improve their flavor, etc. [1]. According to the origins of food materials, fermented foods could be divided into plant sources and animal sources. Plant fermented foods commonly include soy sauce, wine, vinegar, fruit, kimchi, etc., while animal fermented foods commonly include ham, sausage, cheese, yogurt, etc. In the past, researchers paid more attention to the flavor and texture of fermented foods. In the food industry, additionally, it is also expected to obtain fermented products with better flavors and more popular tastes [2]. But as the concept of a healthy diet has taken root, in addition to the flavor and preservation of fermented foods people are also paying attention to the functional activity and health benefits of fermented foods [3]. Many fermented foods are found to have great functional activities, such as hypoglycemic [4], antioxidant [5], hypolipidemic [6], and antihypertension effects [7], and they also improve the human intestinal environment [8], etc. This might be attributed to the possibility that the fermentation processes influences the characteristics of food components [9]. During the fermentation process, some functional microorganisms produce secondary metabolites with anti-pathogenic microbial activities [10]. Moreover, microorganisms could degrade proteins into bioactive peptides [11] and synthesize extracellular polysaccharides with strong biological activities [12,13]. In recent years, researchers have found that polyphenols in plant fermented foods may be the main components conferring their many functional activities [14]. Therefore, the research on polyphenols in plant fermented foods has become increasingly popular.

Polyphenols are a class of plant secondary metabolites with multiple biological activities that are synthesized by plants in the presence of harsh climatic environments or pathogens [15]. As shown in Figure 1, polyphenols are mainly divided into flavonoids and non-flavonoids. Up to now, more than 8000 kinds of phenolic substances have been identified, among which more than 4000 kinds of flavonoids have been identified. Flavonoids are compounds with a parent nucleus structure of two benzene rings connected by three carbon atoms, which can be divided into flavanols, flavones, chalcones, isoflavones, flavanols and flavanones according to their structure. Non-flavonoids include phenolic acids, tannins, astragalus and lignans, etc., mainly referring to a class of substances that must contain at least one aromatic nucleus and more than one hydroxyl group [16]. Phenolic acids are mainly divided into two categories according to their structures, namely hydroxybenzoic acid and hydroxycinnamic acid. Polyphenols are the main bioactive components in fermented plant foods, such as whole-grain fermented foods, vegetable fermented foods and fruit fermented foods. However, only a small amount (5–10%) of the polyphenols in food are absorbed into the body; the unabsorbed substances enter the colon and partially interact with intestinal microorganisms, thus playing a role in promoting the health of the host [17]. Rodríguez-Daza et al. [18] suggested that polyphenols could promote the growth of probiotic bacteria and inhibit the growth of harmful bacteria at the same time. A new term “duplibiotic” was proposed to describe the dual action of polyphenols. In addition, polyphenols promote the secretion of polyphenol-related enzymes (tannase, gallate decarboxylase, esterase, phenolic acid decarboxylase and glycosidase) by intestinal microorganisms to metabolize difficult-to-use phenolics into nutritional growth factors. In a similar way, microorganisms interact with polyphenols during food fermentation. Under the dual action of polyphenols, the quality of fermented foods will be changed. Meanwhile, the hydrolysis of polyphenols by microorganisms could improve the bioavailability of polyphenols in food materials and produce a series of bioactive components that are more favorable for human digestion, absorption and utilization [19].

Although plant foods are rich in polyphenols, there are not enough studies focusing on the overall variation of polyphenols and the changes in their functional activities during fermentation in plant foods. Based on the current research status, the aim of this review is to integrate and analyze the current research on the interaction between polyphenols and microorganisms during polyphenol-rich fermented food (PFF) fermentation. The mechanisms of polyphenol–microbial interactions during PFF fermentation were systematically described from four aspects: (1) changes of functional activity and polyphenols in food after fermentation, (2) polyphenol-associated enzymes (PAEs) secreted by microorganisms, (3) biotransformation pathways of different polyphenols and (4) effects of polyphenols on microorganisms affecting the quality of PFFs. The present study might provide theoretical support for further exploration of the functional activity and bioavailability of PFFs.

## 2. Effect of Microorganisms on Polyphenol Metabolism during PFF Fermentation

Although polyphenols show good biological activity, they still show the drawbacks of a short half-life and low bioavailability after oral administration [20,21]. This may be due to the interaction between the polyphenols and the polysaccharides combined and the formation of glycosides through coulombic interactions, hydrogen bonds, π-π interaction forces, etc., and then the biological activity of the glycosylated polyphenols is reduced. In addition, flavan-3-ol or flavan-3,4-diol form condensed tannins (proanthocyanidins), ellagic acid and gallic acid form hydrolysable tannins, ellagic acid combines with sugar molecules through lipid bonds to form ellagitannins, and gallic tannins and other phenolic polymers have more complex structures and larger molecular weights, which make them difficult to be absorbed after entering the human body, thus reducing their bioavailability. However, during the fermentation process of PFFs, the glycosidic bonds that are contained in large-molecular-weight polyphenols could be metabolized by microorganisms and thus degraded, and the degraded glucoside could be directly utilized by beneficial intestinal microorganisms in the human body as a direct carbon source, thus increasing the abundance of beneficial bacteria. Phenolic compounds can also be metabolized by microorganisms into other small-molecule phenols to enhance their biological activities. During PFF fermentation, key fermentable microorganisms can secrete a variety of PAEs (tannases, esterases, phenolic acid decarboxylases, glycosidases) to release the bound phenols that are contained in a food matrix. This biotransformation induces an increase in the amount and type of free phenols in fermented foods and thus enhances the bioactivity and bioavailability of PFFs. For the details, the mechanism of actions of microorganisms in metabolizing polyphenols during food fermentation to enhance the functional activity of foods are systematically summarized as follows.

### 2.1. Changes of Functional Activity and Polyphenols in PFFs after Fermentation

After spontaneous fermentation or probiotic fermentation, the nutrients contained in PFFs will be fully utilized by microorganisms as substrates, which will result in significant changes in the composition of nutrients contained in PFFs [22]. As one of the important components in PFFs, the type and content of polyphenols also changed significantly. As shown in Abbreviations, all the polyphenols contained in PFFs showed significant differences after fermentation. Generally, the change trend is a transition from large-molecule to small-molecule polyphenols. Macromolecular polyphenols mainly include proanthocyanidins, flavonoid glycosides, ellagitannins, etc. The molecular weight of these polyphenols is usually >500 kDa, and they will be hydrolyzed into small-molecule polyphenols such as aglycone, ellagic acid and catechin by microorganisms. For example, pomegranate, bayberry and other foods are rich in ellagic tannins [23]; during the fermentation, tannase secreted by microorganisms could degrade ellagic tannin to some intermediates (punicalin, gallagic acid), and eventually into ellagic acid. In addition, the content of rutin in food will decrease significantly during fermentation, and the corresponding content of quercetin will increase significantly. This is because rutin is a flavonoid glycoside formed by the combination of quercetin and rutinoside. Under the action of glycosidase secreted by microorganisms, the glycosidic bond contained in rutin is hydrolyzed and broken to form quercetin [24]. In addition, soy is rich in flavonoids, but usually in the form of glucosides. For example, after fermentation, daidzin, glycitin, and genistin are hydrolyzed by probiotics with β-glycosidase activity to form aglycones, such as genistein, daidzein, and glycitein [25]. Moreover, foods rich in proanthocyanidins include blueberry, strawberry, black wolfberry, black mulberry, sorghum, etc. The contents of proanthocyanidins in those food will be significantly reduced after fermentation [26]; this is because anthocyanins such as malvidin-3-glucosides, cyanindin-3-glucoside, peonidin-3-glucoside, and malvidin-3-galactoside could be metabolized to some free phenolic acids (e.g., syringic acid, gallic acid and protocatechuic acid) by probiotics [6]. In addition, foods such as blueberries, blackberries and green coffee beans are rich in chlorogenic acid, which is formed by a lipid bond between caffeic acid and quinic acid. Microorganisms capable of secreting cinnamic esterase can hydrolyze lipid bonds and metabolize chlorogenic acid to caffeic acid and quinic acid during fermentation.

During these fermentation processes, the composition and content of polyphenols in foods are usually significantly changed, and these changes also alter the functional activities of foods, as different polyphenols have different physiological activities or there is a strong or weak gap for the same physiological activity. In general, polyphenols with small molecular mass and unbound glycosides showed stronger physiological activity. However, food fermentation is the process of converting large-molecule polyphenols into small molecules. Therefore, foods normally have higher nutritional value after fermentation. For example, quercetin is a flavonol that easily combines with glycosides to form flavonoid aglycones, such as isoquercitrin and rutin. But in fact, quercetin shows stronger antioxidant, anti-inflammatory activity and an ability to protect the liver [27]. In the fermentation process of sourdough bread [24], black tartary buckwheat [28], jujubes [29] and other foods, rutin and isoquercitrin are all hydrolyzed to quercetin. Moreover, for example, glucosides such as glycitin, daidzin and genistin, which soybeans are rich in, decrease significantly after fermentation, and correspondingly, aglycones such as daidzein, glycitein and genistein increase significantly. Their molecular weights decreased from 416.378 kDa, 432.378 kDa, 446.404 kDa to 284.263 kDa, 254.238 kDa and 270.237 kDa, respectively. Meanwhile, fermented soybeans had higher significant antioxidant activity and better DNA damage protection [30]. In the same way, a similar transformation process was reported for tannins. For example, punicalagin is an ellagic tannin that is abundant in pomegranate, bayberry and other fruits. With a molecular weight of 1084.718 kDa, punicalagin can be bio-converted to ellagic acid with a molecular weight of 302.193 kDa and shows great biological activity. This transformation process usually occurs during the fermentation of fruit juices such as pomegranate juice, bayberry juice and raspberry juice [7]. Moreover, gallic acid (GA), with a molecular weight of 170.12 kDa, is a small-molecule polyphenol with strong antioxidant, anti-inflammatory, anti-hyperlipidemia and other functional activities, and also shows better bioavailability. However, in some foods, GA is usually linked in the form of lipid bonds to form hydrolyzed tannins in some foods, such as green tea, grape, mango, etc. After fermentation, GA could be enriched to enhance the functional activity of food [31]. For example, as a kind of fermented tea, pu’er tea has a better weight loss effect than other teas, and the content of GA is regarded as one of the important indicators to evaluate the quality of pu’er tea [32]. Based on the results of previous studies, more and more researchers are trying to screen microorganisms with better PAE activity to ferment food, to improve the nutritional value of food. As shown in Table 1, some lactic acid bacteria, *Bifidobacteria* and molds have been proved to show strong polyphenol hydrolase activity and were able to be used for the hydrolysis of macromolecular polyphenols.

### 2.2. PAEs Secreted by Microorganisms during Fermentation

#### 2.2.1. Tannase

Tannins are large polyphenolic polymers that form complexes with proteins or other macromolecules (cellulose, starch), mostly composed of lipid bonds among gallic acids, catechins, ellagic acid and glucosidic bonds, with molar masses between 300–3000 Da. The complex structure of tannins gives them a wide range of biological activities, but they are difficult to hydrolyze and absorb by the body. It has been shown that the higher the molecular weight of tannins is, the lower is the bioactivity and bioavailability [46,47]. The anti-nutritional properties of tannins could be reduced by decreasing the tannin polymerization during food fermentation, and the hydrolysis of tannin components by tannases is a feasible method to enhance the digestibility and bioactivity of fermented foods. Tannase, also known as tannin acyl hydrolase, is a functional enzyme produced by microorganisms that, with the ability to hydrolyze hydrolysable tannins, condense tannins, pentosidine tannins and gallic acid lipids into small molecules of phenolics and glucose. Normally, tannase can be produced by fungi [48] and bacteria [49], i.e., yeast, *Lactobacillus plantarum*, *Lactobacillus*, *Bacillus*, and *Aspergillus niger* [50].

As typical condensed tannins, proanthocyanidins (PACs) are pigment components that are widely found in plants. It is a multimer (dimer, trimer up to decamer; below pentamer is called oligomeric PACs, above pentamer is called polymeric PACs) that are formed by the combination of different amounts of catechins and epicatechin [51]. Proanthocyanidins have been reported to have antioxidant [52], antimicrobial [53] and antihypertension properties [54]. However, PACs have low bioavailability after oral administration, and generally PACs with polymerization degree > 4 are not directly absorbed by the body, and small amounts of monomers or dimers could be detected in the plasma [55]. Unabsorbed PACs are eventually fermented and degraded by colonic microflora into small molecular phenolic acids such as hydroxyphenylacetic acid, hydroxyphenylpropionic acid, hydroxyphenylpentanoic acid and 3,4-dihydroxyphenylacetic acid [56]. For example, green tea is a food abundant in PACs, such as (−)-epigallocatechin-3-gallate (EGCG), (−)-epigallocatechin (EGC), (−)-epicatechin-3-gallate (ECG), (−)-epicatechin (EC), gallocatechins (GA) and gallocatechin gallate [57]. Among them, EGCG is the main component, accounting for about 50–70% of the tea polyphenol content [58]. EGCG gives green tea great bioactivity, but the complexity of its structure leads to poor absorption, reduced bioavailability and bioactivity, DNA damage after oral administration [59], and may enhance the incidence of colon cancer [60]. Under the hydrolysis of tannases, EGCG could be hydrolyzed to ECG, EGC, EC and GA, which reduces the potential risk of green tea while increasing its bioavailability and antioxidant activity [61]. For example, Govindarajan et al. [62] investigated the biotransformation of tea samples by tannases produced by *E. cloacae* strain 41 and evaluated the antioxidant capacity of the biotransformed tea. They found that the quality and taste of the tea were improved after tannase treatment. Particularly, the content of ester molecules (EGCG and ECG) in tea liquor also decreased significantly after biotransformation, while the content of non-ester molecules (EGC, EC and GA) and antioxidant capacity increased significantly.

Moreover, blueberry is also a kind of fruit that is rich in PACs, and long-term intake of blueberries is beneficial in preventing cardiovascular disease, type II diabetes and neurological decline [63]. Similarly, grapes are abundant in PACs, and the content and structure of PACs are important factors in the quality of grapes. The fermentation of grapes and blueberries is commonly applied to make wine, but astringency is one of the key factors that affect the sensory characteristics of the wine [64]. The astringency is attributed to the binding of PACs to salivary proteins and oral epithelial proteins, and PACs with a higher degree of polymerization and gallic acyl content showed a stronger astringent taste [65], but gallic acid does not contribute to astringency [66]. This may be due to the more complex structural composition of highly polymerized PACs, which are more likely to bind to salivary proteins to form complexes and thus cause astringency [67], whereas the monomeric flavan-3-ol hardly reacts with salivary proteins [68]. For example, Kyraleou et al. [69] found a positive correlation between astringency and total phenolic concentration, with PACs playing the most influential role. EGCs, monomers and dimers are reduced in astringency because the B-ring is hydroxylated. Therefore, wine showed more astringency as the degree of polymerization of PAC increases.. This outcome is consistent with the studies of Sun et al. [70]. Therefore, adjusting the concentration of these water-soluble tannins during the wine-making process could improve the astringency of less vintage wines [71]. Tannases secreted by microorganisms have been found to reduce the astringency of wines and remove tannin content. For instance, González-Royo et al. [72] added three inactive dry yeast strains for wine fermentation and found a significant decrease in the astringency index as well as a significant decrease in the average degree of polymerization (mDP) of PACs. They found that dry yeast strains alleviated the astringency of the wine by eliminating PACs with high mDP.

#### 2.2.2. Esterase

Hydroxycinnamic acid derivatives are the most abundant polyphenols in cereals. This group includes mainly ferulic acid, caffeic acid, erucic acid and *p*-coumaric acid, which usually exist in the bound state in cereals and are linked to proteins, lignin, cellulose and other cell wall structural components through lipid bonds, forming methyl ferulate, methyl *p*-caffeate, methyl *p*-coumarate, methyl erucic acid and other bound phenols [73]. The bound phenols are not easily digested and absorbed in the body [74]. Esterase is an enzyme with hydroxycinnamate hydrolytic activity secreted by bacteria or fungi *(Lactobacillus gastricus, Lactobacillus acidophilus*, *Bifidobacterium bifidum*, *Aspergillus niger*, *Myceliophthora thermophila*) [75]. Among them, feruloyl esterase (FE) is the most common type of esterase. As shown in Table 2, FEs are classified into four main types, which catalyze the hydrolysis of hydroxycinnamic acid esters in plant cell walls to release hydroxycinnamic acid alone, including ferulic acid [76]. Among the hydroxycinnamic acids that are contained in cereals, ferulic acid is the main and potentially therapeutic substance for treating diabetes, Alzheimer’s disease and other oxidation-related diseases [77]. Currently, strains with the ability to secrete FE are widely used in the solid-state fermentation of cereals and beverages (*Miso*, cake and *Shochu*). For example, Annél Smit et al. [78] found that the esterified bound phenols in grapes are released by cinnamoyl esterases into free phenolic acids (ferulic acid, *p*-coumarin, caffeic acid) during the wine-brewing process, which are then hydrolyzed by microbially secreted phenolic acid decarboxylases into volatile phenols such as 4-vinyl and 4-ethyl derivatives to enrich the aroma of the wine. Zhang et al. [79] used multiple bacteria (*Klebsiella pneumoniae JZE*, *Bacillus mycoides JEF*, *Bacillus cereus JZ3*, *Bacillus velezensis G1*, *Siccibacter colletis G2*, and *Bacillus subtilis strain G6*) in a synergistic manner to ferment waste wine lees (mainly composed of wheat bran and sorghum) left after brewing, which enhances the release of ferulic acid effectively, and the release of ferulic acid from the cell wall of grains by FE makes fermented grain foods more nutritious. In addition, chocolate is a food with a special flavor formed by fermentation of cocoa beans. Cocoa beans are rich in ferulic acid, which has obvious astringency and bitterness [80]. They cannot produce the chocolate flavor by direct baking without fermentation, but after fermentation, ferulic acid is released and the antioxidant activity of cocoa beans is significantly increased, the bitterness is significantly reduced and the flavor is more abundant [81]. The four types of FE (A [82], B [83], C [84,85], D [86]) are listed in Table 2. In addition, microorganisms capable of secreting different types of FE and polyphenols that FE is capable of hydrolyzing are listed in Table 2.

#### 2.2.3. Phenolic acid Decarboxylase

Phenolic acid decarboxylase (PAD) is an enzyme secreted by microorganisms (yeast, lactic acid bacteria, *Mycobacteria*) that converts hydroxycinnamic acid into volatile phenols. PAD transfers protons from the active site to the nucleophilic C2 carbon of the substrate through deprotonation of the phenolic acid hydroxyl group to form intermediates, which promote electron transfer of the ester and lead to C-C bond breakage, resulting in the formation of volatile phenols such as 4-vinyl guaiacol (4-VG) [87]. There are many microorganisms with PAD activity during food fermentation [88]. For example, Svensson et al. [89] fermented four strains of *Lactobacillus* with culture mediums containing caffeic acid, ferulic acid and naringenin-7-O-glucoside, respectively. They found that ferulic acid, caffeic acid and naringenin-7-O-glucoside were all able to be metabolized by *L. plantarum* and *L. ferumum* to small-molecule metabolites (dihydroferulic acid, vinylcatechol, ethylcatechol and naringenin) by PADs. After fermentation, the phenolic acid esters contained in sorghum dough are metabolized to free phenolic acids, and further metabolized to additional metabolites with smaller molecular weights, and improve the nutritional value and taste of fermented sorghum food. A similar conclusion was also reported by Ripari et al. [90], who co-fermented whole wheat dough with *Lactobacillus* and yeast to release the bound ferulic acid and then converted the released ferulic acid into dihydroferulic acid and volatile metabolites; they also suggested that the texture of bread could be improved by targeted conversion of phenolic acids during the fermentation process. Moreover, Xu et al. [91] found that yeast could grow in culture medium containing ferulic acid (50 mg/L) and metabolize ferulic acid to 4-VG, which was an essential flavoring substance in wine, white wine and beer. Rosimi et al. [92] isolated six strains of lactic acid bacteria (*Lactobacillus plantarum* and *Lactobacillus pentosus*) with PAD activities from spontaneously fermented *kimchi*, and they were able to metabolize ferulic acid and *p*-coumaric acid to produce volatile phenols. Mycobacteria are also capable of secreting PADs. For example, Linke et al. [93] found that ascomycete *Isaria farinose,* used to ferment wheat bran and sugar beets, was able to secrete PADs to metabolize ferulic acid to produce 4-VG.

#### 2.2.4. Glucosidase

Polyphenols can be decorated with monosaccharides and are usually combined with monosaccharides such as glucose, rhamnose, fructose, arabinose and galactose [94]. Glycosides are linked to phenolic hydroxyl groups by α-glycosidic bonds or β-glycosidic bonds to form O-glycosides or C-glycosides. O-glycosylation reduces the biological activity of flavonoids, while C-glycosylation enhances the benefits to humans [95]. The glycosylated polyphenols are more water-soluble; their polarity increases with increasing molecular weight, and the chemical structure and degree of hydrolyzability of these compounds determine their bioavailability [96]. Glycosides have more stable structures when combined with sugars to form glycosides, but for the bioavailability, the glycosides need to be hydrolyzed to release the biological activity. For example, C-glycosides show higher biological activity, but they are harder for the body to degrade and utilize than the O-glycosides [97]. And phenols with a single hydroxyl group show very low antioxidant activity when attached to glycosides, probably because the glycosidic bond replaces the hydroxyl group, thus leaving the phenol without a free hydroxyl group to neutralize free radicals [98]. In addition, glycosylation also leads to a reduction in the bioactivity and bioavailability of flavonoids (e.g., glycosylation reduces the antioxidant activity of catechin) [99,100]. Hostetler et al. [101] analyzed the anti-inflammatory activity of celery extracts and found that extracts rich in flavonoid aglycones were effective in reducing the production of inflammatory factors, while the extracts rich in glycosides were not. Additionally, they formulated diets containing glycosides or aglycones to feed mice separately and found that the absorption of aglycones was significantly higher than that of glycosides. 

Glycosidase is an enzyme secreted by microorganisms that can degrade plant cell walls and break glycosidic bonds, and the metabolized glycosides can be converted into carbon sources for microbial growth. During PFF fermentation, glycosidases play a role in improving the flavor and increasing the nutritional function of PFFs. As shown in Table 3, naringin is one of the factors of excessive bitterness in citrus juice. Peng et al. [102] used glycosidase to treat citrus juice and reduced the concentration of naringin from 356.33 mg/L to 10.52 mg/L, which improved the flavor and increased the antioxidant activity. Quercetin, kaempferol and isorhamnetin in sea buckthorn are usually combined with glucose, rhamnose or rutinosek. Gu et al. [103] screened a fungal strain with high glycosidase activity to ferment sea buckthorn, and they found that the total phenolic content of fermented sea buckthorn was significantly increased due to the increased content of flavonol aglycones (quercetin, kaempferol, isorhamnetin). And the antioxidant activity of sea buckthorn was positively correlated with the aglycone content, suggesting that aglycones released by glycosidase hydrolysis of glycosides could increase the antioxidant activity of phenols. Baffi et al. [104] purified a strain of *Bacillus* sp. with glycosidase activity of 20.0 U/mg, which is highly tolerant to ethanol and glucose and could be used for brewing, and they found that the addition of *Bacillus* sp. during wine fermentation could hydrolyze glycosidic terpenes, release free terpenes and enhance the aroma of the wine. Moreover, Na Guo et al. [105] used *Monascus* to solidly ferment mulberry leaves, and found that the glycosides gradually decreased with increasing fermentation time and were almost completely consumed at the end, while at the same time the content of aglycones (quercetin and kaempferol) reached a maximum. They suggested that during the fermentation process, the highly active cellulase broke down the cell walls of mulberry leaves, dissociated the polyphenols in the cell walls of mulberry leaves, and then hydrolyzed the β-glycosidic bonds by β-glycosidase, thus converting the dissociated polyphenolic glycosides into aglycones. Other studies [106,107,108,109] that showed similar results are also summarized in Table 3. 

### 2.3. Biotransformation Pathways of Polyphenols during PFF Fermentation

#### 2.3.1. Biotransformation Pathway of Tannin

The common tannins in food are hydrolysable tannins and condensed tannins, which have different biotransformation pathways. Hydrolysable tannins are composed of glycosidic and lipid bonds. Condensed tannins, also known as proanthocyanidins, are complex polymers with flavan-3-ol as the base structure [110]. As shown in Figure 2A, Ascacio et al. [111] studied the degradation process of prune tannin and found that prune tannin belonged to a polymer of ellagic acid (ellagitannin), in which, under the action of tannase secreted by *Aspergillus niger*, the lipid bond was broken to form punicalin, and under the action of β-glucosidase, the glycosidic bond was cleaved from polysaccharides to form gallagic acid, and finally an ellagic acid monomer was formed. Ellagitannin is also present in pomegranate [23], which could be biotransformed during the fermentation process to enhance its bioavailability [112]. For example, Valero-Cases et al. [113] fermented pomegranate juice with *Lactobacillus* and found a 40–70% reduction in ellagic tannins and a significant increase in ellagic acid content. As shown in Figure 2B, Chavez-Gonzalez et al. [114] used *Aspergillus niger* GH1 solid-state fermentation of tannic acid (pentagalloylglucose, a pentagalloyl tannin) and found that the degree of polymerization of tannic acid decreased sequentially from pentagalloylglucose to tetragalloylglucose, trigalloylglucose and digalloylglucose. This biotransformation is the result of tannic acid breaking the ether bond in the presence of tannase, releasing a gallic group and eventually forming a large amount of gallic acid. Moreover, the gallic acid is further cleaved into pyrogallol, piruvate, and resocinol by the action of descarboxylase, pyrogallol 1, 2 dioxigenase and 5-oxo-6 methyl-hexanoate.

Proanthocyanidins are commonly found in grapes, cocoa, tea and other raw food materials, whose structure is composed of two aromatic rings and a six-membered pyran ring connected by a carbon skeleton, and usually exist in the form of a degree of polymerization > 5 [115]. Its constituent unit is mainly flavan-3-ol and can be divided into type A and type B according to C4-C8 or C6-C8 interflavan linkages. Type A is usually produced by oxidation of type B [116]. As shown in Figure 2D, Wen et al. [117] found that the polymerized proanthocyanidins could be depolymerized into oligomeric proanthocyanidins (dimer, proanthocyanidin B2) during the fermentation process. De Taeye et al. [118] found that proanthocyanidins B2 (C4-C8) contained in cocoa could be converted into proanthocyanidins A2 (C2-O-C7) during baking and fermentation, further formed A2 open structure intermediates, and eventually formed oxidation products (oxidized A-type dimers). In addition, tea polyphenols normally exist in the form of anthocyanins before fermentation. After fermentation, the levels of EGCG and ECG in tea were significantly reduced [119]. In the fermentation of tea, aspergillus is the dominant species and can secrete polyphenol-related enzymes that metabolize tea polyphenols [120]. In the initial stage of fermentation, macromolecular compounds containing ester bonds, such as EGCG and ECG, were mainly present in tea. However, after long-term fermentation, the content of flavanols containing no ester linkage of gallic acid (i.e., EC, GA, and EGC) were gradually increased. Furthermore, under the longer fermentation, these compounds are further degraded into lower molecular substances with simple structures. Qin et al. [121] found that catechin produces an intermediate metabolite of catechin (3, 6-dihydro-6-oxo-2H-pyran moity) during the fermentation process, which is further cleaved into various small molecular substances (Figure 2C). In other words, in the early stage of fermentation, the degradation of gallate bonds was the main pathway for the metabolism of tea polyphenols, but in the later stage of fermentation, the small molecular substances produced by the B-ring biotransformation of flavanols were the reasons for the metabolism of catechins. A similar conclusion was also reported by An et al. [122]. In general, the increase in the biological activity of the fermented tea and the improvement in flavor were both caused by the hydrolysis of the ester bond of tea tannin and the degradation of flavonol glycosides, and finally the formation of GA and other small molecular metabolites, and the contribution of *fungi* (*Aspergillus*) was greater than that of bacteria [32]. In summary, tannin, as a polyphenol with high polymerization degree, exists widely in the PFFs. Under the action of tannase, polyphenol could be hydrolyzed into small molecular substances by means of lipid bond rupture, glycosidic bond rupture, ether bond rupture and oxidation, etc.

#### 2.3.2. Biotransformation Pathway of Phenolic Acid

As shown in Figure 3A, Morata et al. [123] added hydroxycinnamic acid to produce wine by fermentation of grapes with *Saccharomyces cerevisiae*. It was found that the addition of caffeic acid produced two new vinyl phenol pigments (malvidin-3-O-glucoside-4-vinylcatechol and malvidin-3-O-glucoside-4-vinylguaiacol), which is formed by condensation of malvidin-3-O-glucoside and vinylguaiacol formed by decarboxylation of caffeic acid with the action of phenolic acid decarboxylase. Additionally, with the addition of *p*-coumaric acid, ferulic acid could produce five new vinylphenol compounds, and the content of malvidin-3-O-glucide-4-vinyl phenyl was significantly higher than that of other compounds. As the hydroxycinnamic acid in the grape was converted into the vinyl phenol pyran anthocyanin under the action of the esterase, the production of 4-ethylphenol (a negative flavor substance, the excessive content of which can affect the flavor of wine) in wine was reduced and the stability of the wine in the aging process was improved [124]. A similar conclusion was also reported by Vanbeneden Nele et al. [125], who found that 4-vinylguaiacol showed a high content in all kinds of beers (wheat beers, blond specialty beers and dark specialty beers), which was due to the conversion of hydroxycinnamic acid into ethyl derivatives by *Saccharomyces cerevisiae*. Hydroxycinnamic acid in malt is usually acylated with plant cell walls to form the conjugated phenols, and the fermentation process could promote the release of hydroxycinnamic acid and improve the phenolic aroma of wort. Moreover, Ricci et al. [126] found that phenolic acid decarboxylase and phenolic acid reductase secreted by *Lactobacillus* in elderberry juice could metabolize caffeic acid and protocatechuic acid to dihydrocaffeic acid and catechol. Dihydrocaffeic acid and catechol showed stronger antioxidant activity as hydrolysates of hydroxycinnamic acid [127]. As shown in Figure 3B, Sáyago-Ayerdi et al. [128] found that phenolic acids were mainly present in the form of hydroxycinnamates in *Hibiscus sabdariffa L. calyces*. Among them, 5-caffeoylquinic acid, *p*-coumaroylquinic acid and 5-feruloylquinic acid were the main compounds. With the extension of the fermentation time, hydroxycinnamates were hydrolyzed by esterase into hydroxycinnamic acid (caffeic acid, *p*-coumaric acid, and ferulic acid), further metabolized into dihydrohydroxycinnamic acid by phenolic acid reductase, and finally degraded into phenylacetic acid, benzoic acid and its derivatives under the action of decarboxylase. Similarly, after fermentation, hydroxycinnamic acid could form abundant metabolites such as 3-(*ρ*-hydroxycinnamic acid, 3-(3-hydroxycinnamic acid) and 4-hydroxycinnamic acid [129]. Moreover, as shown in Figure 3C, during the fermentation process, *p*-coumaric acid and ferulic acid were easily decarboxylated by phenolic acid decarboxylase, forming metabolites 4-vinylphenol (4-VP) [130] and 4-vinylguaiacol (4-VG) [131]. Further, Kitaoka et al. [132] found that 4-VP and 4-VG could be connected to glucose through ether bonds to form 4-Vinyl Phenyl β-D-glucide (4-VPG) and 4-Vinyl Guaiacol β-D-glucide (4-VGG), and the glucose moiety on 4-VPG and 4-VGG was then connected to xylose through ether bonds to form 4-4-vinylphenol β-primeveroside and 4-vinylguaiacol β-primeveroside (4-VGP), and this biotransformation pathway might be catalyzed by glycosyltransferases [133]. In summary, the bound phenolic acids presented in the food were predominantly hydroxycinnamates, which readily esterified with 1 L-3/4/5- quinic acid to form isomers or derivatives of chlorogenic acid [134]. They could be gradually transformed into ethyl derivatives or small molecular metabolites such as phenylpropionic acid, phenylacetic acid and catechol under the action of esterase, phenolic acid reductase and phenolic acid decarboxylase secreted by microorganisms.

#### 2.3.3. Biotransformation Pathway of Flavonoids

Common flavonoids in plants are mainly divided into four types: flavanol, flavone, flavnone and isoflavone. As shown in Figure 4A, flavonoids (kaempferol, quercetin, myricetin, isorhamnetin, etc.) normally combine with sugar groups (glucose, galactose, rhamnose, xylose, arabinose, etc.) to form glycosides, and the binding mode is that aglycones replace the hydroxyl groups at C-3, C-5, C-7, C-3′, C-4′ and C-5′ of flavonoids to connect with flavonoid aglycones. In particular, the ligands are most easily attached to the C-3 position of flavonoid skeletons to form O-glycosides [135]. Hundreds of glycosides are formed by the connection between the glycoconjugates and flavonoids found in plants. As shown in Figure 4B, catechin is a monomeric flavonoid present in plants, which is often linked to gallic acid by an ester bond to form catechin gallate and is widely found in tea and cocoa [136]. Under the action of tannase and esterase, catechin gallate could be converted into catechin and gallic acid [137], thus improving the absorption capacity and lipid metabolism of the human body [138]. Flavanone is characterized by no double bond in the C2-C3 position of its skeleton structure, like naringenin is the representative. Glucose is easily linked to the C7 position of naringenin, forming naringenin 7-O-glucoside, which could be metabolized by *Lactobacillus plantarum* into naringenin during fermentation [139]. In addition, baicalin, a representative of flavone, is formed by linking glucuronide at the C7 position of baicalein, and a high content of baicalin was found in *Scutellaria baicalensisis*, which has been widely used as a traditional Chinese herbal medicine [140]. Chen et al. [141] used a *Lactobacillus brevis RO1* with β-glucosidase activity to ferment milk containing *Scutellaria baicalensisis* extract and found that the content of baicalein in fermented milk was significantly increased. This indicated that baicalin could be hydrolyzed by fermentation to form baicalein. Moreover, isoflavones (such as genistin, glycitin and daidzin) are widely found in legumes. And it has been proved that isoflavone aglycone can be absorbed by the human body more quickly after taking the same dose of isoflavone glycoside and isoflavone aglycone [142]. During the fermentation process, many microorganisms could secrete esterase, α-glucosidase, β-galactosidase and β-glucosidase to convert isoflavone from glycoside to aglycone [143]. For example, Chen et al. [144] found that the content of isoflavone 7-O-glucoside (genistin, glycitin and daidzin) was significantly decreased while the content of isoflavones (genistein, glycitein, daidzein) was significantly increased in soybean after being fermented by fungi, and the antioxidant capacity of fermented soybean food was significantly improved. Di Gioia et al. [145] also showed a similar conclusion that *Bifidobacterium* could effectively bio-convert bean flavonoids to aglycons in fermented soybean and bean milk. As shown in Figure 4 C, as one of the most common flavanols in plants, rutin is a glycosidic compound formed by connecting quercetin with a glucoside and a rhamnoside, which could be hydrolyzed to a variety of free flavonoid aglycones. For example, Gu et al. [103] isolated a fungus with glycosidase activity from fermented tea and used it to ferment *Hippophae rhamnoides* leaves and found that the content of rutin was significantly reduced after fermentation, while the content of flavanols such as kaempferol, quercetin and isorhamnetin significantly increased. The above results showed that rutin could be bio-transformed into free aglycones through deglycosylation, dihydroxylation, and O-methylation, etc. Rutin in litchi peel could also be deglycosylated by *Aspergillus awamori* fermentation, and this process could be divided into two pathways. In the first, rutin was hydrolyzed to quercetin-3-glucoside and quercetin, and in the second, rutin was further deglycosylated to form kaempferol-3-glucoside and kaempferol after being dehydroxylated to form kaempferol-3-rutinoside [146]. Moreover, Song et al. [147] adopted lactic acid bacteria and cellulase combined to ferment the cactus flavonoid extract (the main flavanols were isorhamnetin-3-O-rutinoside, isorhamnetin-3-O-glucoside, isorhamnetin and quercetin). After fermentation for 24 h, the content of isorhamnetin-3-O-rutinoside decreased to an undetectable level. The content of isorhamnetin-3-O-glucoside increased at the first 12 h and then rapidly decreased but could not be detected after 60 h of fermentation. On the contrary, the contents of quercetin and isorhamnetin were positively correlated with the fermentation duration. The above results indicated that isorhamnetin-3-O-rutinoside could be effectively converted into free aglycones by lactic acid bacteria. In addition, Duckstein, Lorenz and Stintzing [148] pointed out that quercetin and kaempferol were preferentially converted to quercetin and kaempferol; with the prolongation of fermentation time, the C-ring cleavage in the skeletal structures of quercetin and kaempferol, respectively, derived their respective B-ring fission products (protocatechuic acid and 4-hydroxybenzoic acid). Moreover, both quercetin and kaempferol could generate the A-ring fission product phloroglucinol after the C-ring cleavage. Similar results were found in another study by Wang et al. [149] on the transformation pathway of polyphenol contained in fermented red jujube; the authors found that during the fermentation process, the degradation pathway of rutin was as follows: rutin→quercetin-3-O-glucoside→quercetin→taxifolin→3,4-dihydroxyphenylacetic acid→protocatechuic acid→4- hydroxybenzoic acid. In summary, the metabolic pathway of flavonoids in the fermentation process was mainly the conversion from glycosides to aglycones, and this process was mainly catalyzed by the related glycosidases secreted by functional microorganisms. In addition, similar structures of flavonoids may undergo metabolic processes such as dehydroxylation, dihydroxylation and O- methylation.

## 3. Effects of Polyphenols Contained in PFFs on Microorganism

During the PFF fermentation process, microorganisms and polyphenols are in an interactive relationship. Microorganisms metabolize polyphenols into free small molecular substances, and polyphenols selectively promote the growth of certain beneficial bacteria (through hydrolysis of ester bonds and rupture of glycosidic bonds to provide a carbon source for functional microorganisms (lactic acid bacteria, yeast, and *bifidobacterium*) to promote their growth). In addition, polyphenols could inhibit the growth of a variety of pathogenic bacteria (e.g., *E. coli*, *Staphylococcus aureuswine*, etc.), and the inhibition might be achieved by causing the leakage of protein and nucleic acid from pathogenic cells, reducing intracellular ATP concentration, destroying cell walls or cell membranes [150]. In addition, PFFs could profoundly affect the intestinal microflora of the human body after consumption, thereby exerting a positive effect on human health. 

### 3.1. Effects of Polyphenols on Microbial Growth in PFF Fermentation Process

As discussed in the previous sections, many beneficial bacteria (yeast, lactic acid bacteria, and *Bifidobacterium*) can metabolize polyphenols such as tannin, hydroxycinnamic acid, and flavonoid glycoside into small molecular products during the PFF fermentation process. Meanwhile, polyphenols can promote the growth of beneficial microorganisms in the fermentation process and inhibit the growth of pathogenic bacteria, thereby increasing the functional activity and safety of the fermented food [147]. As shown in Table 4, spontaneously fermented sausages are popular with consumers because of their special flavor. However, because there is no leavening agent, the special flavors entirely depend on the microorganisms existing in nature, and the biogenic amine will exceed the limited standard [148]. Zhang et al. [151] fermented rose polyphenol extract with sausage and found that rose polyphenol extract could prevent lipid oxidation of sausages and inhibit the formation of biogenic amines. In addition, rose polyphenol extract could significantly promote the growth of lactic acid bacteria while reducing the total number and diversity of bacteria, thereby improving the safety of fermented sausages effectively. Ahmed et al. [152] adopted a clove extract rich in polyphenol to improve the preservation period of beef burgers and found that clove polyphenol showed high antibacterial activity, inhibited the growth of pathogenic bacteria (disrupting the cell wall and cell membranes of pathogenic bacteria and releasing the contents of the cytoplasm) significantly, and had a broad-spectrum antibacterial effect. Clove polyphenols can also improve the lipid stability of beef burgers during storage. Moreover, Budryn et al. [153] used lactic acid bacteria to ferment legume sprouts to increase isoflavone content. After fermentation, the isoflavone content was increased from 1.1 g/100 g dw to 5.5 g/100 g dw, the number of lactic acid bacteria was also increased by 2 log 10 CFU/mL, and various pathogenic bacteria either did not appear or the content was significantly decreased. Abbasi-Parizad et al. [154] spontaneously fermented the tomato polyphenol extract and found that in the presence of polyphenol, lactic acid bacteria became the dominant species in the natural microflora and accumulated in large amounts, while other pathogenic bacteria died because of the large amount of lactic acid that was secreted by lactic acid bacteria. Li et al. [44] adopted *Lactobacillus plantarum* strains and *Lactobacillus fermentum* strains as the leavening agents for the fermentation of blueberry juice and reported that the number of viable counts of microorganism increased by 35% compared to the control. This is mainly owing to a large accumulation of *Lactobacillus plantarum* and *Lactobacillus fermentum*. Meanwhile, the total phenol content in the blueberry juice was significantly increased, but the proanthocyanidin content was significantly decreased, while the contents of gallic acid, protocatechuic acid and myricetin were significantly increased. Moreover, the antioxidant capacity of blueberry juice was significantly improved. Similar studies [155,156,157,158,159] are also summarized in Table 4. According to those results, we could conclude that polyphenols have a significant promoting effect on the growth of lactic acid bacteria, which could assist lactic acid bacteria to become the dominant species in the microflora. At the same time, lactic acid bacteria could degrade polyphenols such as flavonoid glycosides, proanthocyanidins and hydroxycinnamic acid into free aglycones and small-molecule phenols to increase the total phenol content and biological activity of the PFFs. It is an interactive relationship, which can endow the PFFs with more nutritional functions. Meanwhile, the PFFs could inhibit the growth of pathogenic bacteria in the fermentation process, prevent the accumulation of toxic and harmful microbial secondary metabolites and improve the safety of the fermented food. 

### 3.2. Effects of PFFs on Human Intestinal Microorganisms

Humans are hosts to a large number of diverse microflora, and the distribution of intestinal microflora has a profound impact on human health [159]. The biomass of intestinal microbiota could be as high as 1.5 kg, and more than 1000 species of microbiota have been identified, mainly including *Firmicutes*, *Bacteroidetes*, *Actinobacteria* and *Proteobacteria* [160]. According to their functions, gut microbes can be divided into beneficial bacteria, neutral bacteria and harmful bacteria. Beneficial bacteria mainly include *Lactobacillus* and *Bifidobacterium*, which can inhibit the growth of pathogenic bacteria and decompose, metabolize, absorb and digest nutrients such as polyphenols and active peptides [161]. Harmful bacteria are *Staphylococcus aureuswine*, *Salmonella*, *Campylobacter* and so on. If the balance of intestinal flora is broken, the large growth of harmful bacteria will affect the immune system and lead to a variety of diseases [162]. Neutral bacteria refer to bacteria that are beneficial under normal conditions but can induce diseases once they proliferate in large quantities, such as *Escherichia coli* and *Enterococcus*. A growing number of studies have demonstrated that PFFs can increase the abundance of beneficial gut microbiota. For example, black tartary buckwheat is rich in rutin. Ren et al. [28] used *Bacillus sp. DU-106* to ferment black tartary buckwheat. The content of rutin in the fermented black tartary buckwheat was significantly reduced, and the contents of total phenol, quercetin and kaonferol were significantly increased. The relative abundance of *Lactobacillus*, *Faecalibaculum* and *Allobaculum* in the gut of black tartary buckwheat-fed mice was significantly increased. In addition, Zhou et al. [163] obtained total polyphenol extracts from a fermented tea (Fu brick tea) to study the improvement of intestinal microflora, and found that total polyphenol extracts from a fermented tea could improve intestinal oxidative stress, intestinal inflammation and repair intestinal barrier function in mice. The extracts also improved intestinal microbiota dysbiosis (increasing the abundance of important intestinal microorganisms such as *Akkermansia muciniphila*, *Alloprevotella*, *Bacteroides*, and *Faecalibaculum*). A decrease in these beneficial microorganisms will lead to a decrease in intestinal barrier function and an increase in intestinal endotoxins. Therefore, the restorative effect of PFFs on intestinal barrier functions might be achieved by increasing the number of these beneficial microorganisms. For example, Zhao et al. [164] used unfermented celery juice and probiotic fermented celery juice to conduct dietary intervention in high-fat mice. Compared with unfermented celery juice, probiotic fermented celery juice significantly changed the composition of gut microbiota and hyperglycemic symptoms in mice (increased the relative abundance of *Lactobacillus*, *Faecalibaculum*, *Ruminococcaceae UCG-014* and *Blautia*). Those results suggested that fermented PFFs were more effective in improving human gut microbiota, which was associated with significant changes in polyphenols in PFFs.

## 4. Conclusions

Natural plant-derived foods are rich in polyphenols, but the bioavailability is generally low. The fermented food formed by spontaneous fermentation or probiotic fermentation often shows better flavor, better functional activity and higher bioavailability. On the one hand, during fermentation, beneficial microorganisms could secrete PAEs such as tannase, esterase, phenolic decarboxylase, glycosidase, and metabolize macromolecular-bound phenols such as tannic acid, proanthocyanidins, gallic acid esters, flavonoid glycosides, etc., into free phenols such as quercetin, phloroglucinol, kaempferol, gallic acid, etc. Those small-molecule free phenols showed higher biological activities and bioavailability than the macromolecular-bound phenols. On the other hand, due to the double action of the polyphenols, the growth of pathogenic bacteria is inhibited by the polyphenols during the PFF fermentation process, and functional microorganisms such as lactic acid bacteria, *Bifidobacterium* and yeast, which are selectively promoted to grow, become the dominant strains in the microflora, thereby ensuring the functional activity and safety of the PFFs. After oral administration, some of the PFFs are absorbed by human digestion and metabolism, and some of the PFFs enter the intestinal tract and are used by the intestinal microbial community, thus improving the structure of the intestinal microbial community. Through a series of pathways, PFFs finally showed a variety of functional activities beneficial to human health, such as promoting digestion, resisting oxidation, and lowering blood sugar and blood lipids.

In summary, the current review performed a systematic discussion on the current work and achievements of PFFs from different aspects. According to the existing results, PFFs have good biological activity and bioavailability. However, there are still few studies on the interaction between microorganisms and polyphenols in the PFF fermentation process, and most of the studies mainly focused on the interaction between polyphenols and intestinal microorganisms. In the future, functional microorganisms with high-activity PAEs should be further screened for fermentation of PFFs. Moreover, non-targeted metabolomics and targeted metabolomics should be applied to track the polyphenol cleavage pathway, analyze the changes in phenolic species and content before and after fermentation, and further compare the changes of PFFs biological activity and bioavailability before and after fermentation. In addition, the total phenol content in PFFs did not always increase with the fermentation time. Therefore, we need to pay more attention to the change of polyphenol content during the whole fermentation process to achieve the optimal fermentation time. Meanwhile, we also should optimize the fermentation parameters, so as to realize the maximum use of polyphenols. The exploration of these mechanisms will provide an experimental basis and theoretical support for in-depth exploration of the biological activity of polyphenols and the development of more nutritious fermented foods. 

## Figures and Tables

**Figure 1 foods-12-03315-f001:**
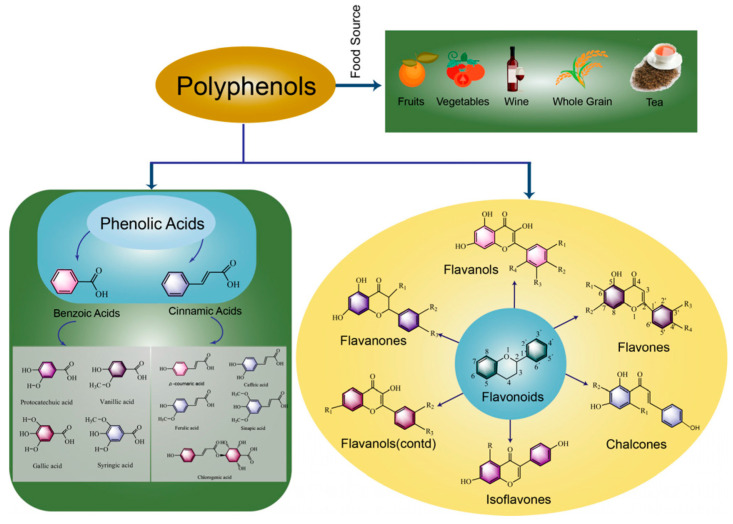
Classification and food sources of polyphenols in fermented foods (polyphenols are classified into two main groups: flavonoids and phenolic acids).

**Figure 2 foods-12-03315-f002:**
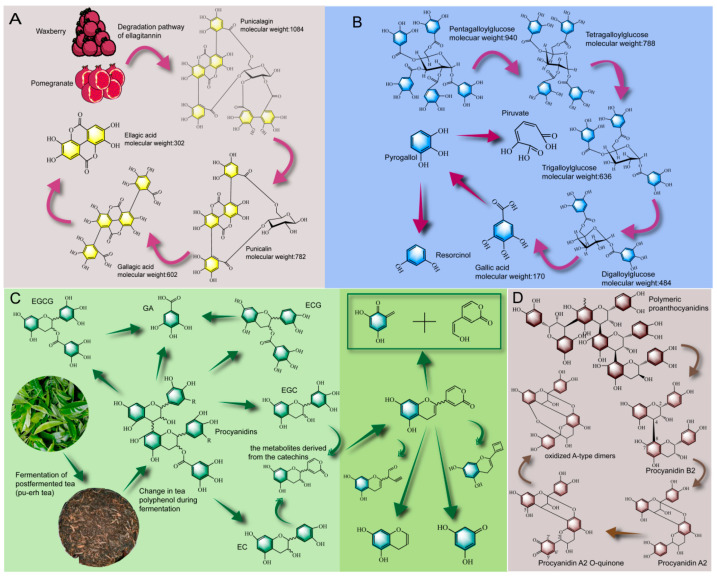
Biotransformation pathways of tannin contained in PFFs: Biotransformation pathway of ellagic tannin rich in *Myrica rubra* and pomegranate (**A**), biotransformation pathway of tannic acid (**B**), biotransformation pathway of tea tannin (**C**), biotransformation pathway of proanthocyanidins (**D**).

**Figure 3 foods-12-03315-f003:**
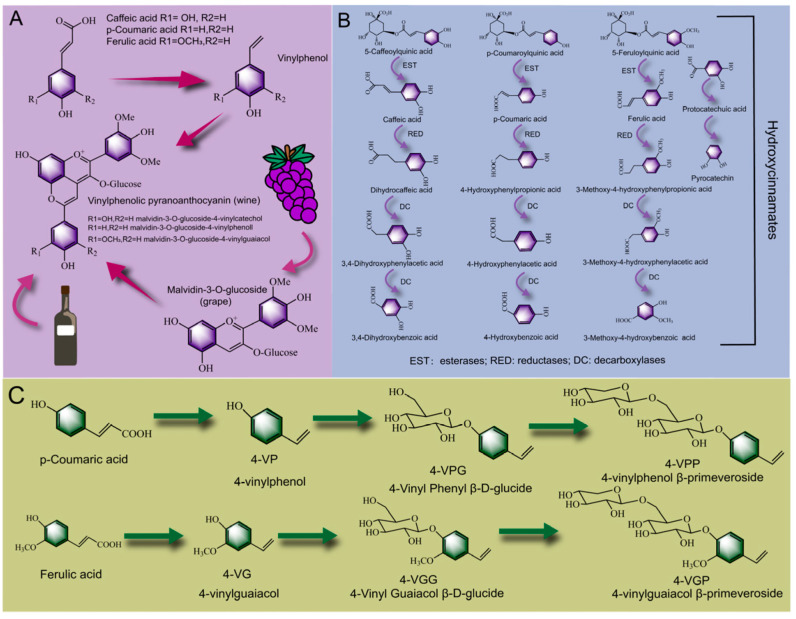
Biotransformation pathways of phenolic acid in PFF fermentation process: the combination pathway of hydroxycinnamic acid and grape anthocyanin in the wine-brewing process (**A**), the biotransformation pathway of caffeic acid ester, ferulic acid ester and *p*-coumaric acid ester (**B**), the biotransformation process of *p*-coumaric acid and ferulic acid into volatile flavor substances (**C**).

**Figure 4 foods-12-03315-f004:**
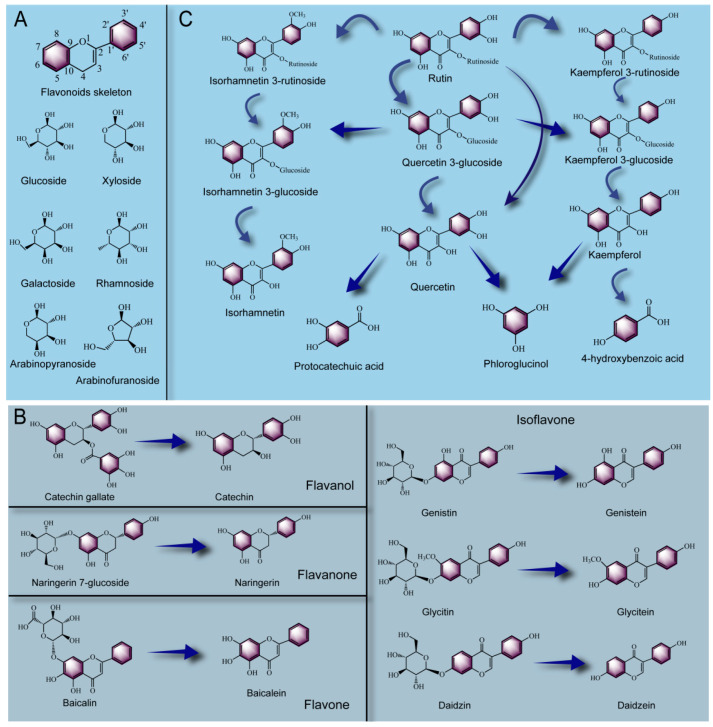
Flavonoid skeleton and its easily linked glycosyl structure producing glycosides (**A**), biotransformation pathway of flavonoids with several different structures (**B**), biotransformation pathway of rutin (**C**).

**Table 1 foods-12-03315-t001:** Changes of polyphenols in PFFs after fermentation. (↑: Increased polyphenol content and functional activity of PPFs; ↓: The content of polyphenols was reduced).

PFFs	Strains Used for Fermentation	Total Phenolic Content	Changes in Polyphenol Content in PFFs after Fermentation	Parameters of Fermentation	Functional Activity Test Doses of PFFs and Corresponding Experimental Models	Changes in Functional Activity	References
Fermented Apulian table olives (*Cellina di Nardò*, *Bella di Cerignola*, *Termite di Bitetto*)	Yeast, lactic acid bacteria	-	Hydroxytyrosol and its derivatives ↑	Healthy olives (150 kg) were collected, washed and placed in plastic vessels of 200 kg capacity filled with 50 L of 12% NaCI (*w*/*v*) for *Cellina di Nardò* and of 10% NaCI (*w*/*v*) for *Bella di Cerignola* and *Termite di Bitetto* cultivars. Olives were co-inoculated with yeast and lactic acid bacteria. Fermentations were carried out at room temperature.	Models: at present, the mainstream antioxidant capacity detection model (DPPH radical scavenging rates) Test dose: DPPH (6~42 μg/mL/Trolox)	Antioxidant activity and bioaccessibility ↑	[33]
Xeniji	Lactic acid bacteria	-	(5-*O*-caffeoylquinic acid, 3-*O*-caffeoylquinic acid, sakuranetin) ↑	-	Models: mice with alcoholic liver injury Test doses: 0.1 g/kg^−1^ BW, 1.0 g/kg^−1^ BW and 2.0 g/kg^−1^ BW	(1) Antioxidant activity ↑, the levels of MDA and ROS in the livers of the Xeniji-treated mice were significantly reduced as compared to the untreated mice. (2) Anti-inflammatory activity ↑, the serum levels of 1 L-1βand IL-6 were significantly lower than in untreated mice. (3) Alcoholic liver injury was significantly improved compared to the untreated mice ↑	[34]
Pomegranate peel extracts (PPE)	*L. acidophilus*	+29.17%	(valoneic acid bilactone, punicalagin, granatin A/lagerstannin A, punigluconin, galloyl-HHDP-hexoside and pedunculagin II) ↑	Sterilized milk (8.6 mL), 0.5–4% (*w*/*v*) PPE, membrane filtered sterilized sucrose solution (0.4 mL) and *L. acidophilus* (1 × 108 CFU/mL, 1 mL) were mixed together and fermented at 37 °C for 0, 4, 8, 12, and 24 h. Milk mixtures without PPE and milk mixtures without adding *L. acidophilus* were used as positive and negative controls, respectively	Models: Trolox-equivalent antioxidant capacity assay Test doses: ABTS value (μmol trolox/mL)	Antioxidant activity ↑, fermented: 27 μmol trolox/mL, non-fermented: 23 μmol trolox/mL.	[35]
Flammulina velutipes	*Bacillus subtilis*, *Bifidobacterium longum*, *Saccharomyces cerevisiae* (Chuangbo microorganism)	+10.11%	(syringic acid, quercetin) ↑	0.10% Chuangbo microorganism, molasses (3%), 28 °C, moisture contents (40%), culture time (10 days)	Models: (DPPH radical scavenging rates, hydroxyl radical scavenging rate, superoxide anion radical scavenging rate and overall reducing power), *RAW264.7* Cells (anti-inflammation capacities)	(1) Antioxidant activity ↑, the DPPH, hydroxyl and superoxide anion radical scavenging rate of fermented foods were significantly higher than in non-fermented foods. (2) Anti-inflammatory activity ↑, the cell viability of the fermented foods group was higher than that of the non-fermented foods group (25, 50, and 100 µg/mL total polyphenols).	[36]
Citrus	*Streptococcus thermophilus, Lactobacillus* *bulgaricus, Bifidobacterium longum, Bifidobacterium breve, Bifidobacterium* *lactis, Bifidobacterium infantis, Bifidobacterium adolescentis, Lactobacillus* *acidophilus, Lactobacillus reuteri, Bifidobacterium bifidum*	-	(Naringin, vanillic acid, gallic acid, dihydrocaffeic acid, phloretic acid, 3-(3-hydroxyphenyl) propanoic acid, 4-hydroxybenzoic acid) ↑ (Hesperidin, neohesperidin, ferulic acid, *p*-coumaric acid) ↓	2 mL samples (0.5 g grapefruit or kumquat or navel orange freeze-dried powder dissolved in 4 mL PBS), 9 mL MRS liquid culture, 9 mL 1% mixed lactic acid bacteria solution, 37 °C, culture time (48 h)	Models: the free radical scavenging of DPPH, ABTS and FRAP and Caco-2 cells Test dose: 5, 10, 20, 25, 30 μg/mL.	Antioxidant activity ↑, the supernatant of fermented citrus had a significant increase in scavenging DPPH, ABTS and FRAP free radical ability.	[37]
Black wolfberry	Spontaneous fermentation	+42.91%	Total anthocyanins ↓, (total flavonoid and polyphenol contents) ↑	Fresh black wolfberry was used to prepare the vinegar, crushed berries and sterile water were used at a ratio of 0.425:1 (*w*/*w*), black wolfberry was fermented in glass jars (28 °C, 60 days), the jars were sealed and placed in a dark temperature-controlled incubator.	Models: (DPPH radical scavenging rates)	Antioxidant activity ↑, the DPPH free radical scavenging ability increased gradually with fermentation time and tended to stabilize from the 20th day, until the 60th day; DPPH free radical scavenging ability reached 63.93%.	[38]
Tartary buckwheat leaves	*Aspergillus niger*	-	Rutin ↓, quercetin ↑	A 680 mL bottle with air-vent capping was used in fermentation, 110 g tartary buckwheat leaves, inoculated with spore suspension (5%, v:m), sterile water (1:1, v:m), 28 °C, culture time: 24 days.	Models: (DPPH radical scavenging rates, Ferricyanide reduction assay)	Antioxidant activity ↑, both the 2 antioxidant activities of fermented foods were promoted, a 150% increase in DPPH scavenging activity and a two-fold higher reducing power.	[39]
Soy sauce (production with soybean and wheat flour)	*Aspergillus oryzae* 3.042	-	(Glucosides, malonylglucosides and acetylglucosides) ↓, (daidzein, glycitein and genistein) ↑	Fermentation process according to China national standard (GB 18186-2000), soybean and wheat flour were mixed at the mass ratio of 4:1 with *Aspergillus oryzae* 3.042 as *koji* for 44 h at 28–30 °C, then *koji* was immersed in a brine containing 17 g/100 g NaCl as *moroml* and fermented for 90 days.	Models: (DPPH radical scavenging rates)	Antioxidant activity ↑, the antioxidant capacity of soy sauce was tripled after fermentation, the maximum value was detected in soy sauce-90 d sample (45.29 ± 0.71 TE μmol/100 mL)	[40]
*Cheonggukjang* (production with soybean)	Spontaneous fermentation	+30.26%	(Daidzin, genisin and glycitin) ↓, (daidzein, glycitein and genistein) ↑	Raw soybeans were washed and soaked for 12 h at 25 °C, then steamed for 45 min at 110 °C and left to stand for 1 h at 25 °C; afterwards, the cooled soybeans were moved to the fermentation room for 72 h at 46 °C, and lastly they were mixed with 4% salt.	Models: at present, the mainstream antioxidant capacity detection model (DPPH radical scavenging rates, ferric-reducing antioxidant power)	Antioxidant activity ↑, the DPPH radical scavenging activity: 99.80 to 104.24 μM TE/g; FRAP values: 38.00 to 63.49 μM TE/g	[41]
Soybean	*Bifidobacterium breve*	-	Glycosides almost disappeared after 24 h of fermentation, while aglycogens increased significantly up to 24 h of fermentation. Aglycones have a better effect than glycosides	Isoflavones were extracted from soybean and added to the medium (400 ppm), the culture was incubated at 37 °C under anaerobic conditions, and samples were removed from each reaction vessel at 0, 10, and 24 h	Models: (1) lipase; (2) mice; (3) 3T3-L1 cells. Test dose: (1) 100 ppm glycosides and 50 ppm aglycones; (2) 1 mg/mL glycosides and aglycones; (3) 0.1, 0.25 mg/mL glycosides and aglycones.	(1) Inhibition of lipase activity ↑, non-fermented 30%→fermented 63.6%; (2) inhibition of adipocyte differentiation and reduction of triglyceride content) ↑	[42]
Fermented soymilk	*Lactobacillus rhamnosus strain ASCC 1520*	-	-	Strain ASCC 1520 was activated by two successive transfers in MRS media, then the activated cultures were transferred into 50 mL of sterile soymilk at an inoculum level of 1% (*v*/*v*) and incubated in a CO_2_ incubator at 37 °C without agitation to obtain fermented soymilk	Models: mice. Test dose: Mice drink freely	(Maintain the balance of intestinal flora, bioavailability) ↑	[43]
Fermented blueberry juices	*Lacto-* *bacillus plantarum, Lactobacillus fer-* *mentum*	+41.8%	(Delphinium-3-glucoside, peonidin-3-glucoside, malvidin-3-glucoside, and malvidin-3- Arabinoside) ↓ (gallic acid, chlorogenic acid, rutin, myricetin and quercetin-3- Rhamnoside) ↑	Blueberries were washed, homogenized and filtered, sterilized to obtain the juices, pH value was adjusted to 4.0 with 1 M Na_2_CO_3_. After pasteurization, juices were inoculated with 1.0% (*v*/*v*) inoculum (7.0 log CFU/mL), 37 °C, 48 h, in darkness.	Models: the free radical scavenging of ABTS and FRAP	Antioxidant activity ↑, non-fermented foods: ABTS and FRAP 68.08 ± 0.25 mmol Trolox/L and 7.22 ± 0.25 mmol Fe^2+^/L; fermented foods: ABTS and FRAP 108.94 ± 0.19 mmol Trolox/L and 12.32 ± 1.77 mmol Fe^2+^/L	[44]
Fermented blueberry juices	*Streptococcus lactis*, *Pediococcus pentosaceus*	-	Malvidin anthocyanins ↓ (catechins, syringic acid, and *p*-coumaric acid, epicatechin, ferulic acid and caffeic acid) ↑	Blueberry pulp was placed into glass bottles (250 mL), fermented blueberry pulp was prepared by inoculating each bottle with 3.2 vol% *Streptococcus lactis* followed by 49 h incubation at 37 °C. (3.7 vol% *Pediococcus pentosaceus* followed by 47 h incubation at 40 °C).	Models: (DPPH radical scavenging rates, ABTS radical scavenging rates).	(1) Antioxidant activity ↑, the ABTS antioxidant capacity of free phenols increased by 8.47%, the DPPH radical scavenging capacity increased by 18.38%; (2) Bioavailability ↑	[45]

**Table 2 foods-12-03315-t002:** Classification of FE.

FE Type	Strain Source	Substrate of Action	Reference
Type A	*A. Tubingensis*, *A. awamori*	Ferulic acid and erucic acid (substrate with methoxy substituents)	[82]
Type B	*N. crassa Fae-1*, *Penicillium funiculosum*	*p*-Coumaric acid, caffeic acid (substrates containing 1 or 2 hydroxyl substituents)	[83]
Type C	*A. oryzae*, *A. niger*, *Talaromyces stipitatus*, *F. oxysporum*	Caffeic acid, erucic acid, ferulic acid, *p*-coumaric acid (four methyl esters of *p*-hydroxycinnamic acid have extensive activity and are more active against *n*-propyl ferulate)	[84,85]
Type D	*Psuedomonas fluorescens* Subsp.	Caffeic acid, erucic acid, ferulic acid, *p*-coumaric acid (broad activity against the four methyl esters of hydroxycinnamic acid and strong hydrolytic activity against acetyl residues)	[86]

**Table 3 foods-12-03315-t003:** Studies related to the hydrolysis of polyphenolic glycosides by glycosidases during PFF fermentation.

Glycosidase Name	PFFs	Target Phenolic Glycosides	Aglycones by Action of Glycosidase	Changes in PFFs Following Glycosidase Action	Microorganisms That Secrete Glycosidase	References
α-L-rhamnosidase, β-D-glucosidase	Citrus Juice	Naringin (356.33 mg/L→10.52 mg/L)	Naringenin	(1) Significantly reduced bitterness, (2) Increased antioxidant activity (DPPH, ABTS radical scavenging ability)	-	[102]
β-glucosidase	*Hippophae rhamnoides* leaves	Rutin (4.61 mg/L→0.92 mg/L)	Quercetin, kaempferol, and isorhamnetin	(1) Increased antioxidant activity (DPPH, ABTS radical scavenging ability and ferric reducing/antioxidant power (FRAP) assay. DPPH: fermented 166.62, non-fermented 124.11; ABTS: fermented 188.32, non-fermented 135.67; FRAP: fermented 212.45, non-fermented 135.67 (mg Trolox equivalents/g dry leaf)) (2) Increased total phenolic content (55.97 ± 1.72 mg GAE/g DW→100.16 ± 3.25 mg GAE/g DW)	*Eurotium amstelodami BSX001*	[103]
Cellulase, β-glucosidase, xylanase, α-amylase	Mulberry leaves	Flavonoid Glycosides were consumed,	Quercetin and kaempferol	(1) Increased antioxidant activity (DPPH and ABTS radical scavenging ability, DPPH: fermented IC_50_ = 95.75 μg/mL, non-fermented IC_50_ = 106.99 μg/mL; ABTS: fermented IC_50_ = 82.19 μg/mL, non-fermented: IC_50_ = 161.58 μg/mL). (2) Increased inhibitory activities of α-glucosidase (fermented: IC_50_ = 35.02 μg/mL, non-fermented: IC_50_ = 95.75 μg/mL). (3) Antibacterial activity (*Candida albicans, Staphylococcus aureuswine*)	*Monascus*	[105]
Cellulase, β-glucosidase, xylanase, hemicellulase	Guava leaf tea	Isoquercitrin, quercetin-3-O-β-D-xylopyranoside, Avicularin, kaempferol-3-O-glucose	Quercetin, kaempferol, gallic acid	(1) Increase in free phenol content (56.1%→88.4%) (2) Increased antioxidant activity (DPPH, ABTS radical scavenging ability, DPPH:fermented IC_50_ = 14.7 μg/mL, non-fermented IC_50_ = 39.5 μg/mL; ABTS: fermented IC_50_ = 4.5 μg/mL, non-fermented: IC_50_ = 9.4 μg/mL). (3) Increased α-glucosidase inhibitory activity (fermented: IC_50_ = 11.8 μg/mL, non-fermented: IC_50_ = 19.2 μg/mL).	*Monascus anka*, *Saccharomyces cerevisiae*	[106]
α-L-rhamnosidase, β-glucosidase	Stingless bee honey	Quercetin-3-rutinoside, Quercetrin-3-rhamnoside	Quercetin and kaempferol	Increased antioxidant ability, antibacterial ability, anti-inflammatory ability and anti-cancer activity	*Penicillium sp.* *Monascus anka, Bacillus* spp.	[107]
β-glucosidase	Black soybeans	Anthocyanins (cyanidin-3-o-β-glucoside), soy isoflavones (isoflavones 7-O-β-d-glucoside)	Genistein, daidzein and cyanidin	Increased antioxidant activity, anti-proliferative activity, and antidiabetic activity	*Rhizoctonia, Bacillus*	[108]
β-glucosidase	Soy milk	Daidzin, genistin	Daidzein, genistein	Increased antioxidant activity (the result of antioxidant activity of soymilks is expressed as a percentage of inhibition; fermented: 83.3%, non-fermented: 20%)	*Lactobacillus*	[109]

**Table 4 foods-12-03315-t004:** Effects of Polyphenols on Microbes in PFF Fermentation.

Polyphenol Composition	Inhibit Pathogenic Bacteria Growth	Promote Probiotic Growth	PFFs	Parameters of Fermentation	Methods for Determining Microbiological Counts	Reference
Hydrolyzable tannin, flavonol	*Eudomonas*, *Psychrobacter*, *Acinetobacter*, *Staphylococcus* and *Kocuria*	*Lactobacillus*	Fermented sausages with rose polyphenol extract	9.6 kg of lean pork and 2.4 kg of backfat were diced into pieces, and supplemented with 1, 2 and 3 mg rose polyphenols per gm meat. Ground meat: 80% lean pork, 20% pork backfat, 2% salt (*w*/*w*), 0.5% sodium glutamate (*w*/*w*), 2% sugar (*w*/*w*), 1% rice wine (*w*/*w*), 0.05% sodium erythorbate, 0.2% sodium polyphosphate(*w*/*w*), 0.012% sodium nitrite (*w*/*w*); blended for 3 min, then stuffed into hog casing; the sausages were fermented, dried in an incubator (20 °C, 90% relative humidity) for 3 days, then 10 °C, 80% relative humidity for 5 days, and 10 °C, 70% relative humidity for 16 days.	Standard plate count method, 16s rDNA sequencing.	[151]
Isoflavonoids	*E.coli*, *Klebsiella sp.*, *Salmonella sp.* and *Shigella sp.*	Lactic acid bacteria	Legume sprouts	50 g of sprouts were flooded with 50 mL of lactic acid bacteria inoculum suspension, fermented condition (30 °C, 48~96 h, depending on the characteristics of the fermented sprouts)	Standard plate count method	[153]
Naringin, kaempferol, gallic acid, hydroxycinnamic acid	-	Lactic acid bacteria	Tomato	Approximately 300 g of tomato (wet weight) were packed into airtight glass containers of 500 mL and pressed to favor the air exit, fluxed with N_2_ before being closed, stored at 20 °C, dark conditions for 240 days.	_-_	[154]
Flavonoid (quercetin, rutin, quercetin, hyperoside, epicatechin, catechin, etc.), phenolic acids (chlorogenic acid, caffeic acid, p-coumaric acid, etc.)	*S.aureus*, *L.monocytogenes*, *E.faecalis*, *E.coli*, *P.aeruginosa*, *C.albicans*	Yeast (*S. boulardii)* *Lactobacilli (L. plantarum)* *L.rhamnosus*	St. John’s wort, winter savory, yarrow, willow gentian	-	Broth microdilution method (results were determined with the minimum inhibitory concentration)	[155]
Syringic acid, ferulic acid, gallic acid	-	*Lactobacillus plantarum*, *Streptococcus thermophilus* and *Bifidobacterium*	Blueberry, blackberry	2% (*v*/*v*) activated potential probiotics were allotted into 100 mL flasks containing 50 mL of pasteurized blueberry or blackberry juice (fermentation 37 °C, 48 h)	Standard plate count method	[156]
Catechin, proanthocyanidin B2, gallic acid, phloretin	-	*Lactobacillus plantarum Y2*	Ginkgo peel	Ginkgo peel juice was pasteurized in a water bath at 90 °C for 20 min; then 1% (*v*/*v*) inoculum of ginko peel juice was inoculated to ensure an initial viable count of approximately 5.0 Log CFU/mL, fermented in an incubator at 37 °C for 48 h.	Standard plate count method	[157]
Daidzein, genistein	-	*Lactobacillus paracasei*, *Leuconostoc mesenteroides*, *Lactobacillus rhamnosus GG* and *Lactobacillus plantarum*	Fermented tofu whey beverage	pH was adjusted to 6 after centrifugation, sterilization at 115 °C for 10 min (tofu whey), then about 500 mL of tofu whey was inoculated with screened strains, then fermented at optimal temperature and pH for 36 h.	Standard plate count method	[158]
Soybean isoflavone	*Staphylococcus aureuswine*	-	Doenjang (the Korean soybean fermented product)	-	Broth microdilution method (results determined with the minimum inhibitory concentration)	[159]

## Data Availability

No new data were created or analyzed in this study. Data sharing is not applicable to this article.

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
