# Peer review of "Effects of Fermentation on Bioactivity and the Composition of Polyphenols Contained in Polyphenol-Rich Foods: A Review"

_foods, 2023, doi:10.3390/foods12173315_

Round 1

Reviewer 1 Report

Dear Authors

Article entitled “Effects of fermentation on bioactivity and the composition of polyphenols contained in polyphenol-rich foods: a review” has been reviewed now. This topic is interesting and authors revealed valuable information about polyphenol compositions and polyphenol rich foods.

However, there are certain points need to be considered for the improvement of the manuscript.

Abstract: A review article is composed of a significant background of the title, which is missing in the abstract. Add background and conclusive lines including with vital polyphenolic components in the abstract.

Introduction: The end part of the introduction should be effective and author should include more

information and justification to review the literature on polyphenol compositions and polyphenol rich foods.

Some other points:

Kindly check the manuscript thoroughly because there are so many typological errors like in some text author use bacterial name starting with uppercase and in some text with lowercase. So, kindly write with uniformity in whole manuscript.

In abstract correct the polyphenols with uppercase i.e., Polyphenols in 3rd line.

Rewrite the name of cheese and wine with uppercase i.e., Cheese and Wine.

In Section 5.1 add the reference no of Jian et al., and Qin et al., ???

In section 5.1 Estefania et al. and Monica L. et al. reference is not found at the end.

In Section 5.2 add the reference no of Antonio Morata et al.???

In Section 5.3 add the reference no of Chen et al.???

At the starting of page no 17 kindly complete the citation of Duckstein, Lorenz & Stintzing

At the starting of page no 18 kindly correct the name of staphylococcus aureus with uppercase i.e. Staphylococcus aureus,

In table no 4 correct the name of bacteria i.e., eudomonas

Article entitled “Effects of fermentation on bioactivity and the composition of polyphenols contained in polyphenol-rich foods: a review” has been reviewed now. This topic is interesting and authors revealed valuable information about polyphenol compositions and polyphenol rich foods.

However, there are certain points need to be considered for the improvement of the manuscript.

Abstract: A review article is composed of a significant background of the title, which is missing in the abstract. Add background and conclusive lines including with vital polyphenolic components in the abstract.

Introduction: The end part of the introduction should be effective and author should include more

information and justification to review the literature on polyphenol compositions and polyphenol rich foods.

Some other points:

Kindly check the manuscript thoroughly because there are so many typological errors like in some text author use bacterial name starting with uppercase and in some text with lowercase. So, kindly write with uniformity in whole manuscript.

In abstract correct the polyphenols with uppercase i.e., Polyphenols in 3rd line.

Rewrite the name of cheese and wine with uppercase i.e., Cheese and Wine.

In Section 5.1 add the reference no of Jian et al., and Qin et al., ???

In section 5.1 Estefania et al. and Monica L. et al. reference is not found at the end.

In Section 5.2 add the reference no of Antonio Morata et al.???

In Section 5.3 add the reference no of Chen et al.???

At the starting of page no 17 kindly complete the citation of Duckstein, Lorenz & Stintzing

At the starting of page no 18 kindly correct the name of staphylococcus aureus with uppercase i.e. Staphylococcus aureus,

In table no 4 correct the name of bacteria i.e., eudomonas

Reviewer 2 Report

Manuscript Title: Effects of fermentation on bioactivity and the composition of polyphenols contained in polyphenol-rich foods: a review

The review presents comprehensive information on the fermentation process of polyphenol-rich food items. It covers extensively the fermentation process mostly involving polyphenols and the micro-organisms involved in the process with attractive pictorial presentation.

 Some minor revisions are pointed out below.

1. Abstract: The first sentence can be rewritten into small more focused sentence. For example, the same can be written as

“Polyphenol-rich fermented foods (PFFs) are reported to have potential health benefits and recognized as main active components. Polyphenols are used as substrates during food fermentation and are hydrolyzed into smaller phenolic compounds with higher bioactivity and bioavailability by polyphenol-associated enzymes (PAEs, e.g., tannases, esterases, phenolic acid decarboxylases and glycosidases). Bio-transformation pathways of different polyphenols by PAEs secreted by different microorganisms are different. Meanwhile, polyphenols could also promote the growth of beneficial bacteria during the fermentation process, while inhibiting the growth of pathogenic bacteria. Therefore, during the fermentation of PFFs, there must be an interactive relationship between polyphenols and microorganisms. A lot of research has focused on the relationship between polyphenols and gut microbes.  The present study is an integration and analysis of the interaction mechanism between PFFs and microorganisms and are systematically elaborated through four aspects, the changes of functional activities of polyphenols in food after fermentation, introduction of PAEs that secreted by functional microorganisms, the bio-transformation pathways of different polyphenols during fermentation and the last part mainly focuses on the effect of polyphenols on microbial growth during fermentation and the interaction between PFFs and gut microbes. The present study will provide some new sights to explore the functional bioavailability of polyphenols in polyphenol-rich fermented foods”.

2. Introduction Line 7 ‘And in the food industry’ better replace with ‘In food industry’

3. Introduction 2nd para, Line 1: ‘The food fermentation process is a complex biological transformation process’ better replace with ‘The food fermentation is a complex biological transformation process’.

4. In section 4.1, Line 5: ‘It has been shown that the higher molecular weight of tannins, the lower bioactivity and bioavailability’ should be corrected as ‘It has been shown that the higher is the molecular weight of tannins, the lower is the bioactivity and bioavailability’

5. Page 3: “Rodríguez Daza et al. suggested that polyphenols could promote the growth of probiotic bacteria and inhibit the growth of harmful bacteria at the same time” – Please add Ref no.

6. In Figure 1: Please include ‘Tea’ also as a food source of polyphenols because tea contains around 20-25% polyphenolic compounds.

7. In Figure 1, the names of benzoic acid and cinnamic acid derivatives are not clear. Please increase the dpi value of the figure, if possible.

8. A list of abbreviations used before the introduction will be very helpful for the reader.

9. “For example, as a kind of fermented tea, Pu 'er tea has a better weight loss effect than other teas, and the content of GA is regarded as one of the important indicators to evaluate the quality of Pu 'er tea [41]” – Please correct the name of the tea type here.

10. Please define the upward and downward arrows in the table footnote.

11. Page 7: Section 4.1: “Tannins are large polyphenolic polymers that form structurally complexes with proteins………” I think the author may remove the word “structurally” from this part.

12. Page 8: “Proanthocyanidins have been reported to have the abilities of antioxidant [60], antimicrobial [61] and hypotensive [62].” – I think one word is missing after ‘hypotensive’.

13. Page 8: “…..PACs with a higher degree of polymerization and gallic acyl content showed a stronger astringent taste……” is it gallic acyl or gallic acid?

14. “For instance, González-Royo et al. added three inactive dry yeast strains for wine fermentation and found a significant decrease in the astringency index as well as a significant decrease in the average degree of polymerization (mDP) of PACs and the ability of them to alleviate the astringency of the wine was achieved by eliminating PACs with high mDP [79]” – Please make the necessary correction with the reference cited. Ref 79 is not by González-Royo et al.

15.  Page 9: “For example, Annel et al. found that the esterified bound phenols in grapes are released by cinnamoyl esterases into free phenolic acids (ferulic acid, p-coumarin, caffeic acid) during the wine brewing process, which are then hydrolyzed by microbially secreted phenolic acid decarboxylases into volatile phenols such as 4-vinyl and 4-ethyl derivatives to enrich the aroma of the wine [85]” -  Please make the necessary correction with the reference cited here.

16. Page 11:  I would suggest adding the reference numbers in the paragraph “Glycosidase is an enzyme……… results were also summarized in Table 3.”

17. Page 12: “As shown in Figure 2 A, Jian et al. studied the degradation process of prune tannin and found that prune tannin belonged to a polymer of ellagic acid (ellagitannin), which under the action of tannase that secreted by Aspergillus niger, the lipid bond was broken to form punicalin, and under the action of β-glucosidase, the glycosidic bond was cleaved from polysaccharides to form gallagic acid, and finally ellagic acid monomer was formed [118].” Please check the reference.

18. Page 12:  “For example, Estefania et al. fermented pomegranate juice with Lactobacillus and found a 40-70% reduction in ellagic tannins and a significant increase in ellagic acid content [120]”- Please make the necessary correction with the reference cited here.

19. Page 12: “As shown in Figure 2 B, Monica L. et al. used Aspergillus niger GH1 solid-state fermentation of ………………………….. and 5-oxo-6 methyl-hexanoate [121]” – Please correct the reference.

20. Page 13: “Taeye et al. found that proanthocyanidins B2 (C4-C8) contained in cocoa could be converted into proanthocyanidins…….” Please check the reference

21.  Please check the “pyruvate” word in Fig 2B.

22. Page 114: “As shown in Figure 3A, Antonio Morata et al. added hydroxycinnamic acid to produce wine by fermentation of grapes with Saccharomyces cerevisiae.” - Please make the necessary correction

23. Page 114: “A similar conclusion was also reported by Nele Vanbeneden et al. [132]………” – Please remove the first name of the author

24. Page 15: “As shown in Figure 3B, Ayeri et al. [135] found……” make the necessary correction to the author name

25. Figure 3B- please increase the dpi value for clarity

26. Figure 3C- I suggest defining the abbreviated names like that in Fig 3B

27. Page 16: “As shown in Figure 4 B, Catenin is a monomeric flavonoid present in plants….”- This is catechin, not catenin.

28. Page 18: “Parizad et al. naturally fermented the tomato polyphenol extract and found that in the presence of polyphenol, lactic acid bacteria became the dominant species in the natural microflora and accumulated in large amounts, while other pathogenic bacteria died because the large amount of lactic acid that was secreted by lactic acid bacteria [164].” _ please correct the author’s name

29. Table 4: First column heading “polyphenol composition”- Change the first letter to uppercase

30. Page 20: “In addition, Zhou et al. [167] obtained total polyphenol extracts………..” please correct the author’s name

31. page 20: “For example, Zhao et al. [170] used unfermented celery juice and probiotic fermented…..” Ref 170 is not Zhao et al. – Please check

32. Figure 5 is not discussed anywhere in the text. I suggest incorporating this figure in the text.

33.  Finally I suggest authors check the references thoroughly.

Reviewer 3 Report

Minor spelling errors were encountered. Carefully check the English language style.

``But a lot of research has focused on the relationship between polyphenols and gut microbes, and to the best of our knowledge, few studies have been focused on this.`` - Unclear phrase.

``Thus, the present studies are systematically integrated and analyzed, and the interaction mechanism between PFFs and microorganisms are systematically elaborated through four aspects, the first is changes of functional activities and polyphenols in food after fermentation, the second is the introduction of PAEs that secreted by functional microorganisms, the third is the bio-transformation pathways of different polyphenols during fermentation, and the last part mainly focuses on the effect of polyphenols on microbial growth during fermentation and the interaction between PFFs and gut microbes.`` - this part is much to long and needs to be rephrased in order to be clear and impactful. What about the impact to the food industry of these products? 

`` Plant fermented foods commonly include soy sauce, white wine, vinegar, fruit wine, kimchi, etc`` - delete ``white``

``Numerous studies have focused on 2 the flavor changes of food after fermentation. And in the food industry, it is also expected to obtain fermented products with beĴer flavor and more popular taste`` - You said numerous studies but you cite only one. The phrase is unclear though.

``But up to now, the beneficial effects of fermented foods on human health are not clear.`` - What do you mean? Please rephrase and include the different perspectives - food industry and health - of the functional food concept.

Replace ``More and more`` with other words.

``The food fermentation process is a complex biological transformation process and could be divided into natural and artificial fermentation`` - What do you mean to artificial? This classification is not a good one. ``Natural`` - you meant spontaneous? I guess so. Please reformulate. 

``Therefore, if we want to control the formation of target fermentation products more precisely and inhibit the production of pathogenic bacteria and toxic substances in food products, more researches on the interactions between various components contained in food raw materials and the microbiota involved in the fermentation process is urgently needed, and to gain insight into the metabolic pathways of various components in the fermentation process, the formation pathways of biologically active fermentation products and the process of beneficial microorganisms forming the dominant strains in the fermentation flora.`` - A lot of research is already available. Please rephrase this part and refer to the recent literature on this topic.

Table 1 - The terminology ``brewing microorganisms`` is strangely and improperly used. Please rephrase. In this table you have to include more information such as: bioactive compounds responsible of each functional activity, the dose from which that product is reported as functional, the recommended daily dose of that food product, fermentation type with parameters - all with the supporting citing references

Eliminate the abbreviation ``PFF`` and include it in the manuscript, starting with the Introduction section where is currently missing.

Section 3 - you specified in this section is about the changes during fermentation, but it should not only be about the enumeration of bioactive compounds, but the factors influencing the fermentation process too. Clearly specify the temperatures impact, protection methods, advanced green technology to increase the polyphenol content prior to and after the fermentation. The journal is not a health related one and you should point out more the technological aspects, all related to food industry. The audience of journal Foods (as an Open access one) is mainly composed on food industry actors interested in improving their food process to increase both the technological and health-related functionality. So in this review you have to stay on this way. 

Tables 1 and 2 can be unified.

Table 3 is too simple, unclear and incomplete. As mentioned before, you have to include concentrations, parameters, etc

Carefully adjust the table numbering in the entire manuscript.

All the Latin denominations must be written with Italics.

Section 7 - once again clear values for concentrations, fermentation parameters are missing.

What is the role of Figure 5 in the conclusion section? The conclusion section should not contain additional new information but specify the main ideas treated in the manuscript. 

As in its current structure, the manuscript is not properly structured to be published as a review paper. Is seems more as a draft paper that still needs a lot of improvements. 

The English language needs improvement.

Round 2

Reviewer 3 Report

My recommendation for this paper is to be published after a careful check for the English language. There are still some minor spelling errors.

My recommendation for this paper is to be published after a careful check for the English language. There are still some minor spelling errors.